# SERPINH1 functions as a multifunctional regulator to promote the malignant progression of cervical cancer

Qian Liu[1,2☯], Yuanhao Peng[2☯], Wenbin Liu[3*], Xiangjian Luo[1,2,4*]

1 Hunan Key Laboratory of Oncotarget Gene, Hunan Cancer Hospital and the Affiliated Cancer Hospital of Xiangya School of Medicine, Central South University, Changsha, Hunan, PR China, 2 Key Laboratory of Carcinogenesis and Invasion, Chinese Ministry of Education, Cancer Research Institute, School of Basic Medicine, Central South University, Changsha, Hunan, PR China, 3 Department of Pathology, Hunan Cancer Hospital and The Affiliated Cancer Hospital of Xiangya School of Medicine, Central South University, Changsha, Hunan, PR China, 4 Key Laboratory of Biological Nanotechnology of National Health Commission, Central South University, Changsha, Hunan, PR China

☯ These authors contributed equally to this work.
* luocsu@csu.edu.cn (XJL); liuwenbin@hnca.org.cn (WBL)

## Abstract

Cervical cancer remains the second leading cause of female cancer mortality worldwide, with metastasis representing a critical therapeutic challenge. This study systematically reveals the key role of SERPINH1 (Serpin Family H Member 1) as a hub regulator of malignant progression in cervical squamous cell carcinoma and endocervical adenocarcinoma (CESC). Through analysis of TCGA-CESC datasets, we identified that high SERPINH1 expression is significantly correlated with poor prognosis and contributes to tumor progression by promoting cell proliferation, invasion, and metastatic phenotypes. In vitro experiments validated these findings, demonstrating that SERPINH1 overexpression markedly enhanced the proliferation, invasion, and metastasis of cervical cancer cells, whereas its knockdown substantially inhibited these processes. Furthermore, based on the SERPINH1-related differentially expressed genes, a prognostic risk model was constructed, successfully identifying PLOD1, ITGA5, and ESM1 as core collaborative genes affecting patient prognosis. Overall, our findings underscore the multiple functions of SERPINH1 as a hub for cervical cancer metastasis regulation, suggesting its potential as a promising biomarker for tailoring strategies in metastasis patients of CESC.

## Introduction

Cervical cancer is the second most common malignancy among women globally, exceeded only by breast cancer, and is especially prevalent in low- and middle-income countries, where it ranks among the top five causes of cancer-related incidence and mortality [1,2]. Recent epidemiological studies highlight the profound

**Data availability statement:** All data generated or analysed during this study are included in this published article and its Supporting Information files.

**Funding:** This work was supported by grant from the Fundamental Research Funds for the Central Universities of Central South University (2025ZZTS0140). The funder had no role in study design, data collection and analysis, decision to publish, or preparation of the manuscript.

**Competing interests:** No conflicts of interest, financial or otherwise, are declared by the authors.

impact of maternal cancer deaths on child welfare: in 2020, an estimated 1 million children worldwide were orphaned due to maternal cancer-related deaths, with nearly half of these cases attributed to breast and cervical cancers [3]. Metastasis is a leading cause of cervical cancer mortality [4,5], primarily driven by stromal invasion, lymphatic spread, and hematogenous dissemination [6]. A critical mechanism underlying tumor metastasis is the Epithelial-Mesenchymal Transition (EMT), through which cancer cells acquire enhanced invasive and migratory properties [7].

SERPINH1 (Serpin Family H Member 1), also known as heat shock protein 47 (HSP47), is a member of the serine protease inhibitor (serpin) family [8]. It serves as a collagen-specific molecular chaperone, playing a critical role in collagen metabolism by facilitating the synthesis, folding, and secretion of type I and III collagens. SERPINH1 specifically binds to nascent collagen precursors in the endoplasmic reticulum, ensuring proper triple-helix formation and stability, thereby maintaining extracellular matrix (ECM) homeostasis and contributing to tissue fibrosis [8,9]. Its functions in embryonic development, wound healing, and fibrotic diseases have been well-documented [9].

In recent years, SERPINH1 has emerged as a key player in tumor biology, with its dysregulated expression implicated in the malignant progression of various cancers [10]. In breast cancer, esophageal squamous cell carcinoma, and gastric cancer, elevated SERPINH1 levels promote collagen deposition and remodeling of the tumor microenvironment (TME), enhancing tumor cell invasion and metastasis [11–14]. In colon cancer, SERPINH1 has been shown to induce chemoresistance through activation of the AKT signaling pathway [15]. Additionally, SERPINH1 regulates tumor cell death mechanisms, including autophagy and apoptosis, further underscoring its multifaceted role in cancer biology [16]. Clinical analyses have consistently demonstrated that high SERPINH1 expression correlates with increased lymph node metastasis risk and poorer survival outcomes in patients with liver cancer, head and neck cancer, and clear cell renal cell carcinoma [17–19].

Despite these advances, the specific functions and regulatory mechanisms of SERPINH1 in cervical cancer development and progression remain poorly understood.

## Materials and methods

### Patient data sets

The clinical and molecular data (including mRNA expression and copy number) of primary cervical cancer patients were retrieved from The Cancer Genome Atlas (TCGA) database through the R package TCGAbiolinks [20]. We downloaded gene expression quantitated as fragments per kilobase of transcript per million mapped reads (FPKM) and Masked Copy Number Segment data generated by Affymetrix SNP 6.0 array. Moreover, we also downloaded the survival information from TCGA Pan-Cancer Clinical Data Resource [21]. Since the TCGA-CESC database contains transcriptomic sequencing data from only six adjacent normal samples, we further selected two independent cervical cancer datasets (GSE9750 and GSE63514) from the GEO database (Gene Expression Omnibus; http://www.ncbi.nlm.nih.gov/gds) for differential expression analysis.

Regarding prognostic analysis, we performed survival analysis using the survival package in R (version 4.3.1). Specifically, the survfit function was employed to construct Kaplan-Meier survival curves. The log-rank test was used to assess the statistical significance of survival differences between the SERPINH1 high-expression and low-expression groups, with simultaneous calculation of hazard ratios (HR) and their corresponding 95% confidence intervals. Pan-cancer survival analyses of genes were performed using the GEPIA2 database (http://gepia2.cancer-pku.cn) [22]. Cox proportional hazard ratio and significance heat map were obtained from this website.

## Cell culture

We used the Hela, Siha, and C33A cell lines, all of which were obtained from the Cancer Research Institute of Central South University. Among these, SiHa and C33A are representative of squamous cell carcinoma, while HeLa is an adenocarcinoma cell line, providing a broader understanding of the SERPINH1 effects. Additionally, our preliminary Western blot validation showed that SERPINH1 expression was relatively highest in HeLa cells, while it was lower in SiHa and C33A cells. This guided our decision to knock down SERPINH1 in HeLa cells and overexpress it in SiHa and C33 cells, aiming to induce greater variations in SERPINH1 expression levels within these cell lines, thereby highlighting the impact of this molecule on cancer cells. These cells were cultured in DMEM medium supplemented with 10% fetal bovine serum (FBS) and maintained in a humidified incubator at 37°C with 5% $CO_2$.

## Differential expression analysis and LASSO regression analysis

We applied the "limma" package for the differential analysis. Differential expression genes (DEGs) was determined by identifying genes with fold change of ≥1 and adjusted P value (FDR) of ≤0.05. We used least absolute shrinkage and selection operator (LASSO) Cox regression to reduce the dimensionality and select the most robust markers to construct the riskScore features. The optimal value of the penalty parameter λ was determined by 10-fold cross-validation.

## Functional enrichment analysis

GSEA and gene set variation analysis (GSVA) were performed with the R package "clusterProfiler" [23] based on gene sets of MsigDB. Functional gene expression signatures (Fges) were applied for an overview of the profile of immune characteristics [24].

## Primers and plasmids

The SERPINH1 overexpression plasmid was purchased from Youbio Biotechnology Co., Ltd. The siRNA targeting SERPINH1 was obtained from Sai Suofei Biotechnology Co., Ltd. The sequences were as follows:

SiSERPINH1#1:GGCACUGCGGAGAAGUUGATT;UCAACUUCUCCGCAGUGCCTT.
SiSERPINH1#2:GCUCAGUGAGCUUCGCUGATT;UCAGCGAAGCUCACUGAGCTT

PCR-SERPINH1-F TGCTAGTCAACGCCATGTTCT
PCR-SERPINH1-R ATAGGACCGAGTCACCATGAA
PCR-PLOD1-F AGACCAAGTATCCGGTGGTGT
PCR-PLOD1-R CTTGAGCACGACCTCATCCAA
PCR-ITGA5-F GGCTTCAACTTAGACGCGGAG
PCR-ITGA5-R TGGCTGGTATTAGCCTTGGGT
PCR-ESM1-F ACAGCAGTGAGTGCAAAAGCA
PCR-ESM1-R GCGGTAGCAAGTTTCTCCCC

## Western blotting

Cells were harvested and washed three times with PBS, then disrupted in IP lysis buffer (25 mM Tris-HCl, pH 7.4, 150 mM NaCl, 1% NP40, 1 mM EDTA, 5% glycerol; Thermo Scientific, MA, USA). Extracted proteins were quantified and

separated by SDS-PAGE and transferred onto nylon membranes. Before incubation with the primary antibody, the nylon membrane was cut based on the size of the target molecules, retaining the bands around 37 kDa for incubation with the GAPDH antibody and approximately 46 kDa for SERPINH1 antibody incubation. After overnight incubation with the primary antibody, peroxidase-conjugated secondary antibodies were used to detect binding of primary antibodies. Visualization was performed by using the ChemiDoc XRS system with Image Lab software (Bio-Rad, CA, USA). The primary antibodies used were GAPDH (Cat#60004–1-lg, Proteintech, Wuhan, China) and SERPINH1 (Cat#10875–1-AP, Proteintech, Wuhan, China).

### RNA extraction and quantitative real-time polymerase chain reaction (q-PCR)

Total RNA was extracted using the Nucleozol reagent (Macherey-nagel GmbH & Co, Düren, Germany) according to the protocol established by the manufacturer. Reverse transcriptional PCR was performed using the RevertAid First Strand cDNA Synthesis kit. The qPCR analysis was conducted in a 7500 Real Time PCR System (Applied BioSystems) using the SYBR Green PCR Supermix (Thermo Fisher Scientific). The PCR reaction conditions were 10 s at 95°C followed by 40 cycles of 15 s at 95°C and 60 s at 60°C. Each sample was examined in triplicates.

### Cell proliferation, plate colony-formation assays, migration and invasion

In a 96-well plate, approximately 800–1200 cells were seeded per well. The CCK8 assay kit was used to measure absorbance values at 450 nm at 0 h, 24 h, 48 h, 72 h, and 96 h to plot the proliferation curve. For the colony formation assay, approximately 500–1,000 cells per well were seeded in a 6-well plate. After visible colonies formed, the cells were fixed with methanol and stained with 0.1% crystal violet, followed by counting the number of colonies. For the cell migration assay, $5 \times 10^4$ cells per well were seeded in a 24-well Transwell plate. After cell migration, the upper chamber cells were removed, and the lower membrane-bound cells were fixed, stained, and counted. In the cell invasion assay, 80 µL of 10% Matrigel was added to the upper chamber of a 24-well Transwell plate, and $4 \times 10^4$ cells per well were seeded. After 36 hours of treatment, the upper chamber cells were removed, and the invasive cells on the lower membrane were fixed, stained, and quantified. All experiments were repeated at least thrice.

### Statistical analysis

Statistical analysis were analyzed using the R software (V.4.3.2, R Core Team, Foundation for Statistical Computing, Vienna, Austria) or GraphPad Prism 9 software (GraphPad Software Inc.). Student's t-test was used for variables that met the requirements for normal distribution, but for non-parametric data the Mann-Whitney U is used. Survival curves were estimated with the Kaplan-Meier method and subsequently compared using log-rank tests. P-value was set at $p < 0.05$ indicates significance. For all analysis, two-by-two pairs indicate statistically significant differences. *, **, *** and **** indicate, respectively $<0.05, <0.01, <0.001$, and $<0.0001$.

## Result

### Result 1 SERPINH1 is upregulated in cervical cancer patients and affects prognosis

Through differential expression analysis of cervical cancer GEO datasets GSE9750 and GSE63514, we identified a significant upregulation of SERPINH1 expression in tumor tissues compared to normal controls ($p < 0.001$) (Fig 1A and 1B). Kaplan-Meier survival analysis further demonstrated that patients with high SERPINH1 expression exhibited significantly worse overall survival compared to those with low expression (hazard ratio [HR] = 2.51, 95% confidence interval [CI]: 1.56–4.02, $p = 0.000077$) (Fig 1C). Notably, SERPINH1 expression did not show significant changes in precancerous lesion samples from the GSE63514 dataset (Fig 1D), suggesting that its dysregulation is specific to malignant tumor phenotypes rather than early precancerous stages. Additionally, SERPINH1 expression levels were significantly positively

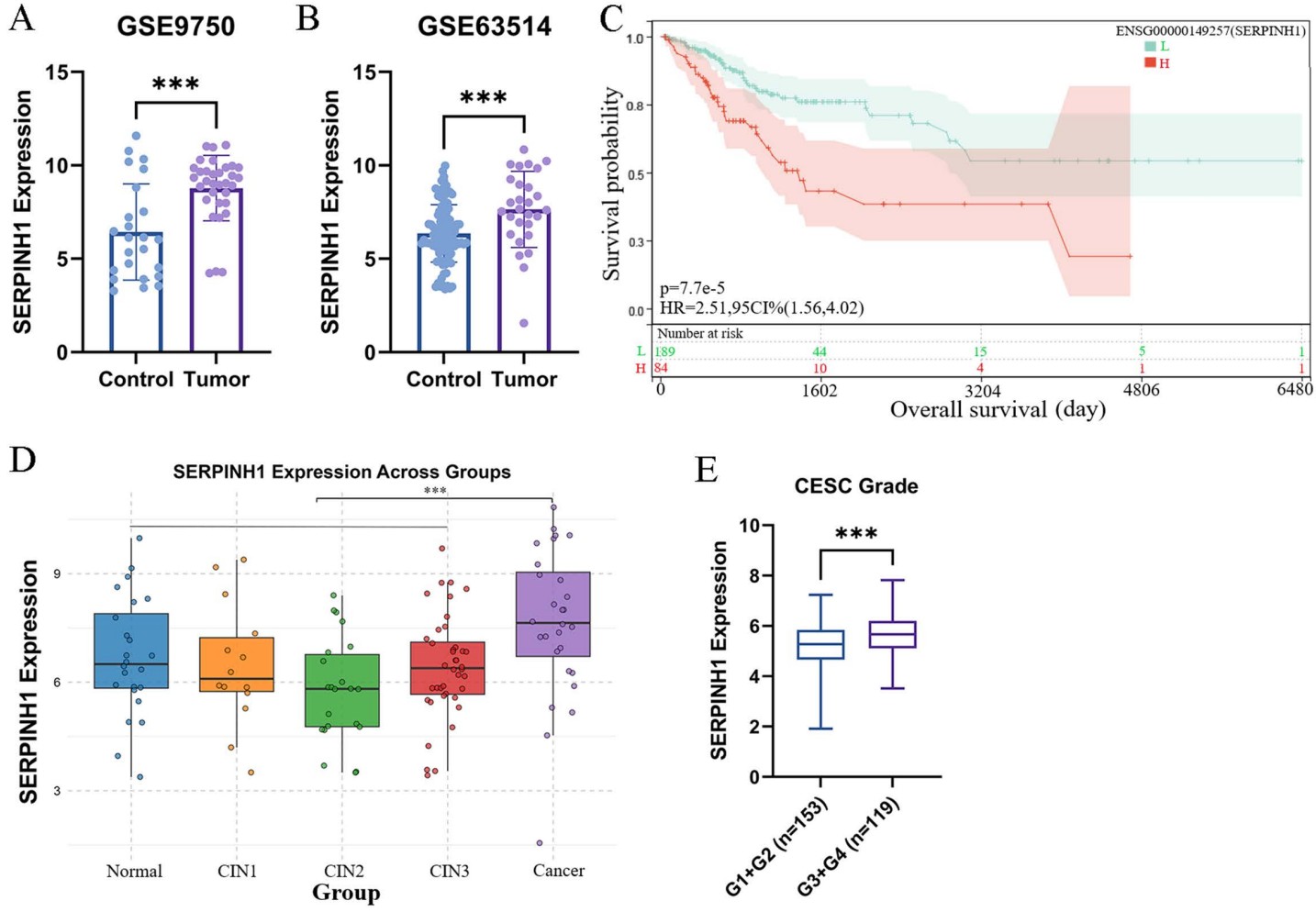

**Fig 1. SERPINH1 is upregulated in cervical cancer patients and affects prognosis.** A-B SERPINH1 expression in cervical cancer tissues compared to normal controls. C Survival curves for SERPINH1 in CESC patients. D SERPINH1 expression in precancerous lesion samples from the GSE63514 dataset. E SERPINH1 expression levels in different histological grade of cervical cancer patients. CESC, Cervical squamous cell carcinoma. (* $P<0.05$; ** $P<0.01$; *** $P<0.001$).

correlated with the histological grade of cervical cancer (Fig 1E). Taken together, these findings suggest that SERPINH1 is likely involved in the malignant progression of cervical cancer and influences patient prognosis, rather than playing a role in tumor initiation.

### Result 2 SERPINH1 is associated with invasion and metastasis phenotype

To investigate the functional role of SERPINH1 in cervical cancer, we performed a genome-wide correlation analysis using the TCGA-CESC dataset (Fig 2A). KEGG pathway enrichment analysis of genes significantly positively correlated with SERPINH1 revealed their primary association with key signaling pathways, including Focal adhesion, ECM-receptor interaction, PI3K-Akt signaling, TGF-beta signaling, and Protein processing in the endoplasmic reticulum (Fig 2B). These pathways are critically implicated in tumor invasion and metastasis. Further validation using Gene Set Enrichment Analysis (GSEA) demonstrated that SERPINH1 positively correlated genes were significantly enriched in tumor invasion

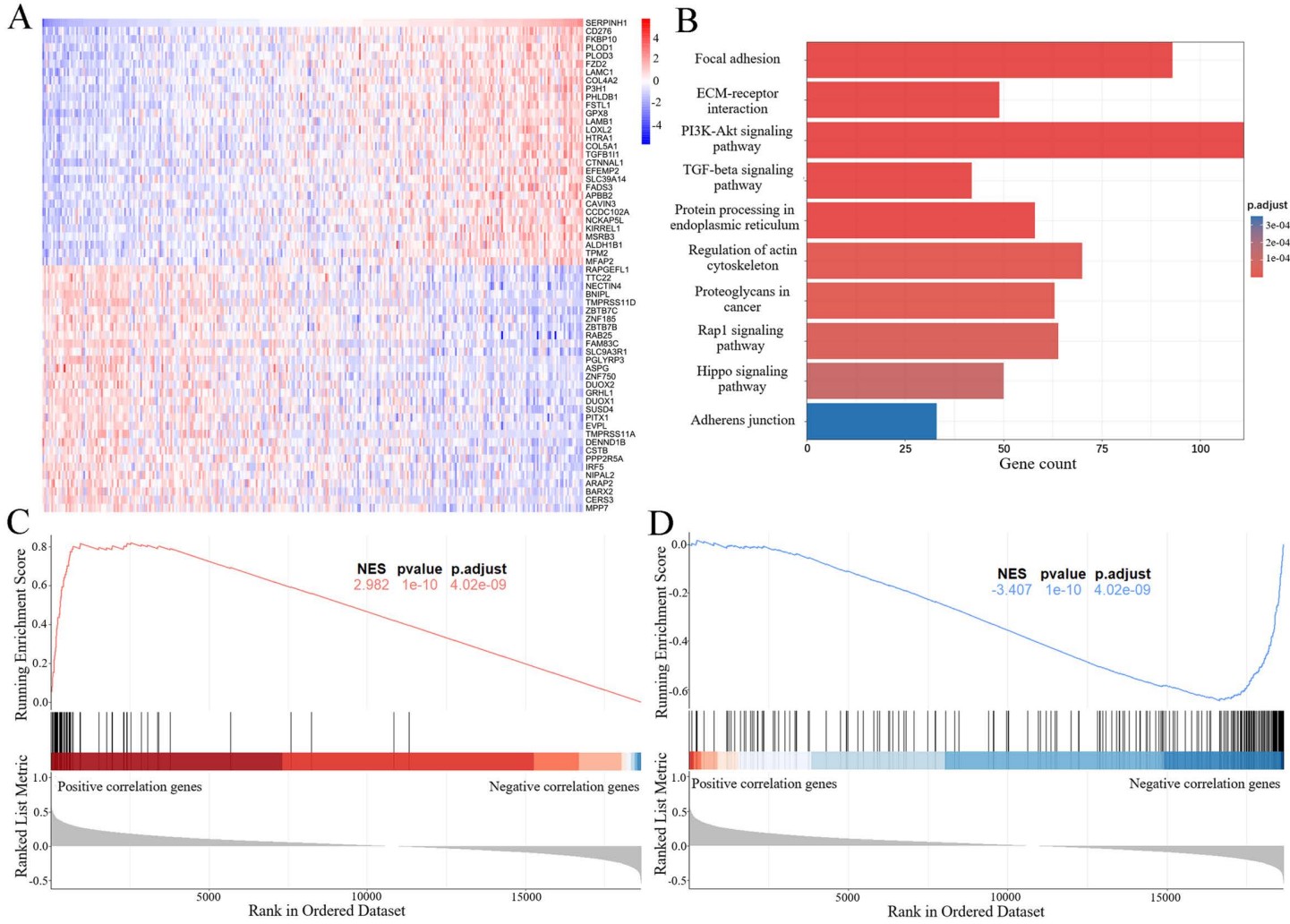

**Fig 2. SERPINH1 is associated with invasion and metastasis phenotype.** A Heatmaps of the top 50 genes that show positive and negative correlations with SERPINH1 in the CESC cohort. B Enrichment analysis of SERPINH1 related genes by KEGG pathways. C-D Gene set enrichment analysis (GSEA) demonstrates the difference between positively and negatively correlated genes of SERPINH1 in (C) Anastassiou multicancer invasiveness signature (D) Jaeger metastasis dn pathways.

pathways (Fig 2C), whereas its negatively correlated genes were prominently enriched in metastasis suppression pathways (Fig 2D). These findings suggest that SERPINH1 plays a significant role in promoting invasion and metastasis in cervical cancer.

### Result 3 Representative biological characteristics of CESC patients with different SERPINH1 expression levels

To evaluate the biological characteristics and immune status of CESC patients, we employed the Gene Set Variation Analysis (GSVA) method, stratifying patients based on high and low SERPINH1 expression. Initial GSVA analysis utilizing the Hallmark [25] gene sets revealed significant differences in multiple biological features (S1 Table). The high-expression group demonstrated pronounced activation of gene sets associated with invasion, metastasis, proliferation promotion, and

angiogenesis. Notably, compared to the low-expression group, the high-expression group exhibited widespread metabolic suppression, including downregulation of gene sets related to glycolysis, hypoxia, fatty acid metabolism, and adipogenesis (Fig 3A).

Further analysis using the BIOCARTA database to evaluate pathway activation in the two patient groups (S2 Table). The results demonstrated that the differential pathways between SERPINH1 high-expression and low-expression patients were predominantly associated with cell migration and metastasis, growth and proliferation, metabolism and energy production, as well as immune and inflammatory responses (Fig 3B). Consistent with these findings, analysis based on the KEGG_LEGACY database further validated that the SERPINH1 high-expression group exhibited enhanced activity in pathways related to invasion and metastasis (S3 Table; Fig 3C).

Notably, the analysis provided several key insights into the metabolic regulatory role of SERPINH1. First, SERPINH1 appears to broadly suppress lipid metabolism, including pathways associated with fatty acids, linoleic acid, pantothenate and CoA, and arachidonic acid, while selectively promoting the biosynthesis of unsaturated fatty acids (Fig 3C). This suppression of lipid metabolism may represent a critical mechanism underlying the invasion and metastasis of CESC cells, highlighting its role in the regulation of nutrient metabolism. Second, the analysis revealed that patients in the SERPINH1 high-expression group displayed significantly upregulated activity in glycan metabolic pathways, encompassing both glycan synthesis and degradation (Fig 3C). These findings suggest that SERPINH1 may play a multifaceted role in modulating metabolic processes that contribute to tumor progression.

Based on previous studies, SERPINH1 has been implicated in tumor progression through the regulation of immune functions [26] and cell death processes, such as apoptosis and autophagy [16]. To further elucidate the functional mechanisms of SERPINH1, we adopted the research methodology of Bagaev A et al. [24] and utilized Functional Gene Expression Signatures (Fges) to evaluate the functional states of various cell populations within the Tumor Microenvironment (TME). These Fges represent the major functional components of tumor, immune, stromal, and other cell populations.

Our analysis revealed that, compared to the SERPINH1 low-expression group, the high-expression group exhibited significant angiogenesis and fibrosis characteristics, while no notable differences were observed in other functional aspects (S4 Table; Fig 3D). This finding suggests that SERPINH1 may promote tumor invasion and metastasis by enhancing angiogenesis and fibrosis. Furthermore, we investigated the relationship between SERPINH1 and twelve cell death patterns [27,28]. The results demonstrated that in SERPINH1 high-expression patients, signature genes associated with Netotic cell death (NETosis) were significantly activated, whereas those related to Alkaliptosis were markedly suppressed (S5 Table). Interestingly, the expression of genes involved in apoptosis and autophagy did not show significant changes (Fig 3E). These findings suggest that SERPINH1 may influence CESC progression by modulating specific cell death modalities, such as NETosis and Alkaliptosis, rather than through classical pathways like apoptosis or autophagy. This highlights a novel mechanism by which SERPINH1 contributes to tumor biology and underscores its potential as a therapeutic target.

## Result 4 SERPINH1 overexpression promotes proliferation, migration and invasion of CESC cells

Based on previous analyses suggesting the tumor-promoting role of SERPINH1, we evaluated its impact on the malignant phenotypes of cervical cancer cells through functional experiments. Initially, we selected two cervical cancer cell lines, Siha and C33A, and established stable overexpression models of SERPINH1 via lentiviral transduction (Fig 4A). The Cell Counting Kit-8 (CCK-8) assay revealed a marked increase in cell viability in the SERPINH1-overexpressing cells compared to the vector control group after 72 hours of culture (Fig 4B and 4C). Subsequent colony formation assays confirmed a significant elevation in the colony formation rate for both cell lines with SERPINH1 overexpression (Fig 4D and 4E). Moreover, Transwell assays demonstrated that the migratory and invasive capacities of cells with SERPINH1 overexpression were substantially enhanced relative to the control cells (Fig. 4F–4I).

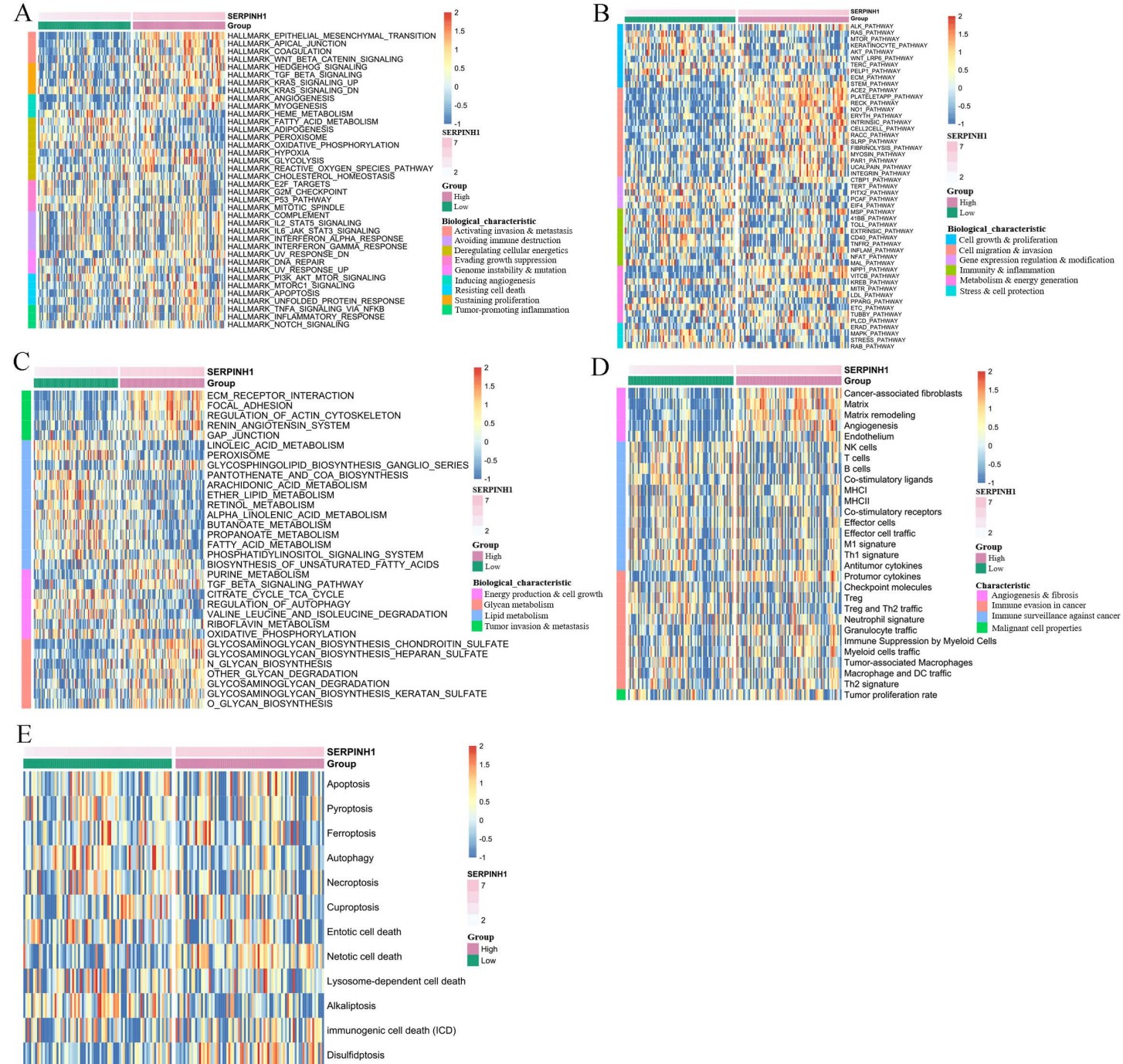

**Fig 3. Representative biological characteristics of CESC patients with different SERPINH1 expression levels.** A The features of hallmarks between low- and high- SERPINH1 expression groups. B Biocarta-based biological characteristics between low- and high- SERPINH1 expression groups. C KEGG-based biological characteristics between low- and high- SERPINH1 expression groups. D The evaluation of Fges for TME. E Evaluation of the relationship between SERPINH1 and twelve cell death patterns. Fges, functional gene expression signatures; TME, Tumor microenvironment.

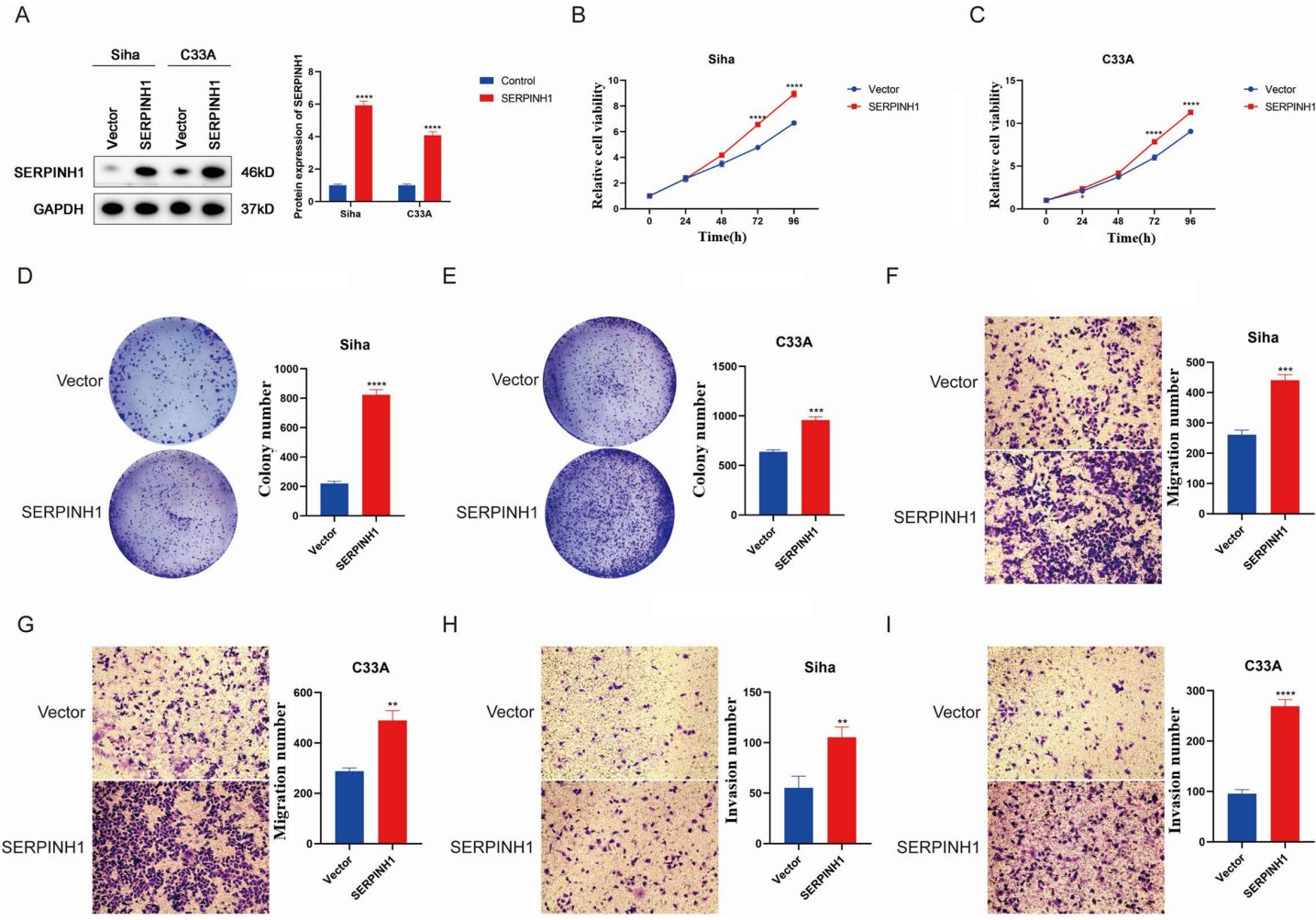

**Fig 4. SERPINH1 overexpression promotes the proliferation, migration and invasion of CESC cells.** (A) The protein level of SERPINH1 in Siha and C33A cells was analyzed by Western blot. (B, C) The overexpression of SERPINH1 promotes the proliferation capability of CESC cells. (D, E) The overexpression of SERPINH1 promotes the clonogenicity of CESC cells. (F, G) The overexpression of SERPINH1 promotes the migration ability of CESC cells. (H, I) The overexpression of SERPINH1 promotes the invasion ability of CESC cells. Each experiment was conducted at least three times. (* $P < 0.05$; ** $P < 0.01$; *** $P < 0.001$, **** $P < 0.0001$, ns, no significance).

### Result 5 SERPINH1 knockdown suppresses the proliferation, migration and invasion of CESC cells

To verify the functional specificity, we knocked down SERPINH1 in the Hela cell line using siRNA transfection, and the knockdown efficiency was confirmed by Western blot (Fig 5A). The CCK-8 assay showed that the proliferation rate of the knockdown group was significantly reduced compared to the control group (p < 0.001) (Fig 5B), and the colony formation ability was weakened (p < 0.001) (Fig 5C). Consistent with these findings, Transwell assays demonstrated that the migration and invasion abilities of the knockdown group were significantly decreased (Fig 5D and 5E). The knockdown and overexpression studies of SERPINH1 confirmed that SERPINH1 regulates the malignant phenotypes of cervical cancer, promoting its proliferation, migration, and invasion.

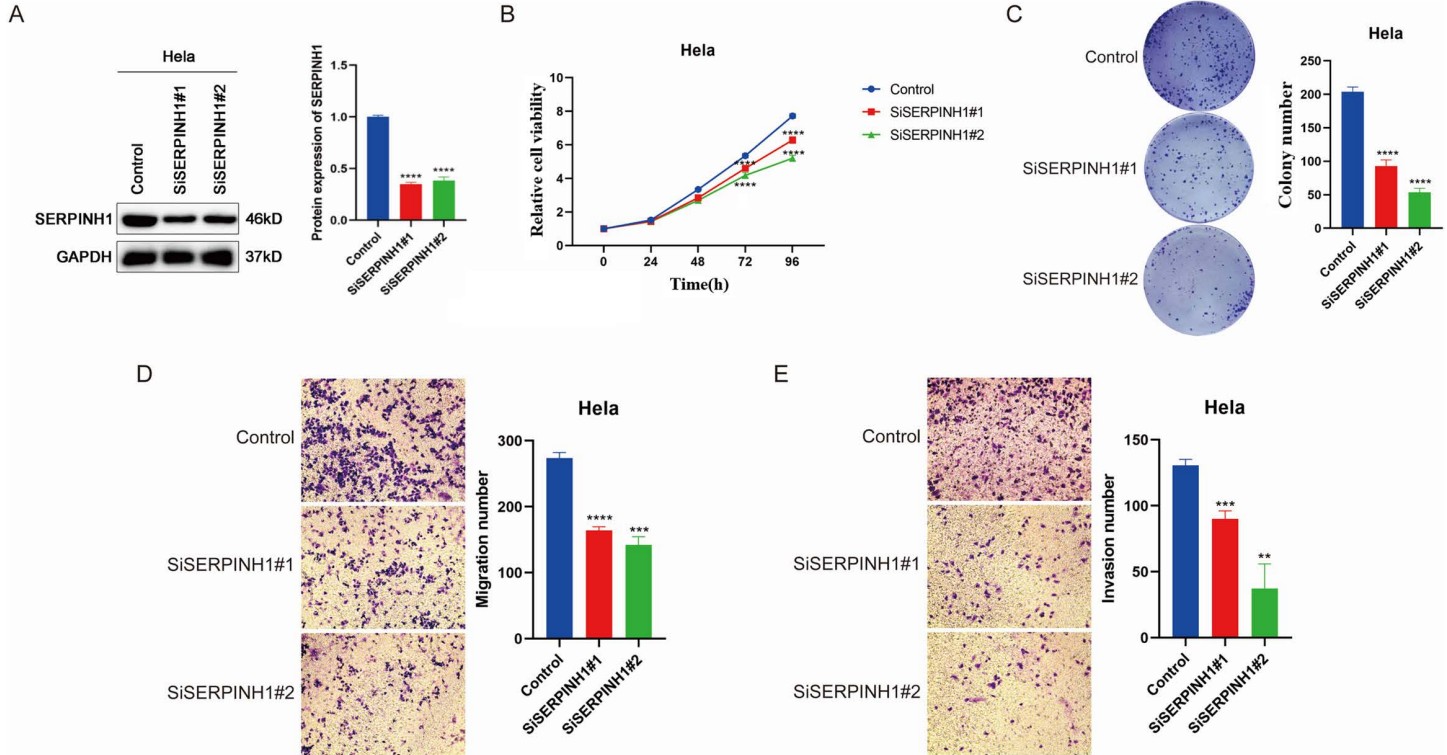

**Fig 5. SERPINH1 knockdown suppresses the proliferation, migration and invasion of CESC cells.** (A) The protein level of SERPINH1 in Hela cells was analyzed by Western blot. (B) The absence of SERPINH1 suppresses the proliferation capability of CESC cells. (C) The absence of SERPINH1 inhibits the clonogenicity of CESC cells. (D) The absence of SERPINH1 inhibits the migration ability of CESC cells. (E) The absence of SERPINH1 inhibits the invasion ability of CESC cells. Each experiment was conducted at least three times. (* $P < 0.05$; ** $P < 0.01$; *** $P < 0.001$, **** $P < 0.0001$, ns, no significance).

### Result 6 SERPINH1-PLOD1-ITGA5-ESM1 gene sets play a synergistic role in influencing prognosis

To further provide valuable information for clinical prognosis, we identified core genes that may influence the prognosis of CESC patients in cooperation with SERPINH1. Transcriptomic differential analysis of patients with high and low SER-PINH1 expression revealed 427 significantly differentially expressed genes (|log2FC|>1, FDR<0.05) (Fig 6A). KEGG enrichment analysis demonstrated that these genes were enriched in key biological processes, including cell adhesion, migration, and EMT (Fig 6B). From these, we selected the top 100 genes based on the significance of differential expression for LASSO regression analysis, ultimately identifying PLOD1, ITGA5, and ESM1 as core genes that influence patient prognosis (Fig 6C–6E). In CESC patients, PLOD1, ITGA5, and ESM1 exhibited significant positive correlations with SERPINH1 (Fig 6F–6H). Using qPCR, we validated changes in the mRNA levels of PLOD1, ITGA5, and ESM1 in Hela cell line with SERPINH1 overexpression or knockdown (Fig 6I). To evaluate the pan-cancer relevance of this gene set, we extended our analysis to 33 cancer types in the TCGA database. The results revealed that the SERPINH1-PLOD1-ITGA5-ESM1 gene set was significantly associated with overall survival of patients with CESC, KIRP (kidney renal papillary cell carcinoma), LGG (brain lower grade glioma), and MESO (mesothelioma) (Fig 6J). This suggests that this molecular network may play analogous roles in these malignancies, highlighting its potential as a broadly relevant prognostic and therapeutic target across multiple cancer types.

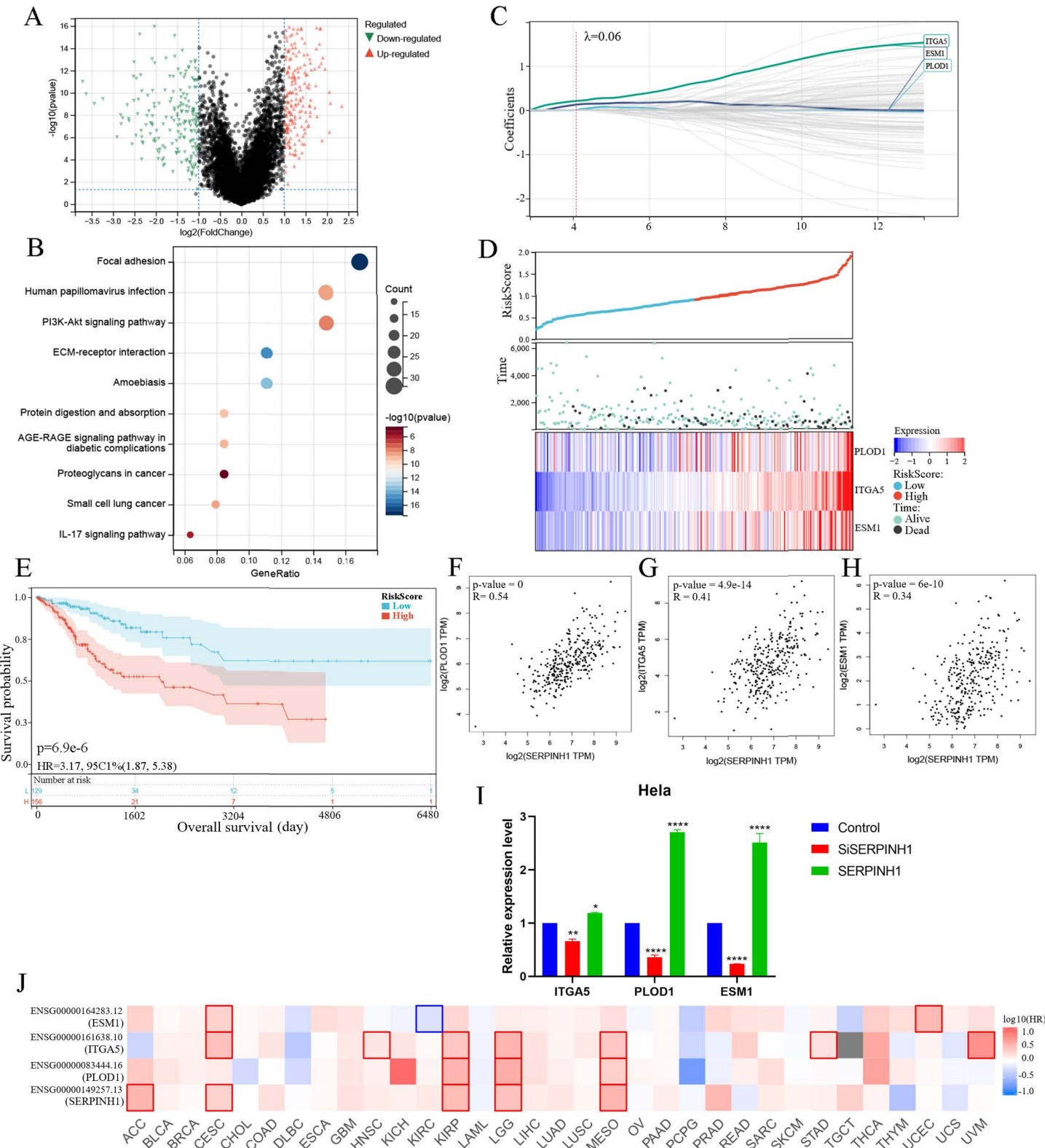

**Fig 6. SERPINH1-PLOD1-ITGA5-ESM1 gene sets play a synergistic role in influencing prognosis.** A Differential analysis based on the expression of SERPINH1. B Enrichment analysis of differential genes by KEGG pathways. C LASSO analysis of differential genes. Adjustment of parameter

selection by tenfold cross-validation in the LASSO model. D Risk score and heatmap of prognostic biomarker expression in CESC patients based on LASSO modeling. E Survival analyses using Kaplan-Meier curves for low- and high- risk score groups. F-H Correlation analysis of expression between SERPINH1 and PLOD1 (F), ITGA5 (G) and ESM1 (H) respectively. I Detection of mRNA levels of PLOD1-ITGA5-ESM1 in Hela and its SERPINH1 knockdown and overexpression cell lines. Each sample was examined in triplicates. J Prognostic analysis of SERPINH1-PLOD1-ITGA5-ESM1 in pan-cancer. LASSO, Least Absolute Shrinkage and Selection Operator.

## Discussion

This study reveals the critical role of SERPINH1 in CESC and its underlying mechanisms. The findings demonstrate that SERPINH1 expression levels are significantly correlated with the prognosis of CESC patients and influence disease progression by regulating malignant phenotypes, including cancer cell proliferation, invasion, and metastasis. Further analysis indicates that the TME of patients with high SERPINH1 expression exhibits pronounced stromal remodeling and angiogenesis, coupled with enhanced EMT activity. In vitro functional experiments further corroborate that SERPINH1 significantly promotes the proliferation and invasion capabilities of CESC cells. These findings are consistent with previous research reports, such as the association of abnormal SERPINH1 expression with various vascular diseases [29–31]. In low-grade glioma [32] and renal cell carcinoma [14,33], SERPINH1 is involved in extracellular matrix remodeling, thereby promoting tumor metastasis.

It is noteworthy that our analysis also uncovered several intriguing findings. CESC patients with high SERPINH1 expression exhibited widespread suppression of lipid metabolism, yet selectively activated the unsaturated fatty acid synthesis pathway, which may be an adaptation to the metabolic demands of tumor metastasis [34]. Studies have shown that the metastatic process is a biologically inefficient event, requiring cancer cells to undergo metabolic reprogramming to adapt to nutritional restrictions, hypoxic stress, and immune surveillance within the circulatory system. During this process, cancer cells often transition from a synthetic to a catabolic metabolic state to sustain their viability during metastasis [34]. There is a well-established, complex interplay between lipid metabolism reprogramming and tumor metastatic phenotypes [35–37]. Of particular interest, polyunsaturated lipids have been demonstrated to support the extravasation and colonization processes of cancer cells during metastasis [38]. These findings revealed an inverse correlation between SERPINH1 expression and activity scores of lipid metabolic pathways. While this suggests a potential link, direct mechanistic evidence for SERPINH1-driven lipid reprogramming requires experimental validation. The clarification of this mechanism could offer novel molecular targets for the development of specific therapeutic strategies aimed at combating metastatic CESC.

Our study revealed that SERPINH1 may significantly enhance glycan metabolism. Given the critical role of glycan modification in tumor progression—including cell signaling, invasion, angiogenesis, and metastasis [39], this finding provides a novel mechanistic perspective on oncogenic activity of SERPINH1. Notably, we identified a unique regulatory relationship between SERPINH1 and two emerging cell death modalities: NETosis-related genes were markedly upregulated, while alkaliptosis-associated genes were suppressed in SERPINH1 high-expression CESC patients. This contrasts with prior reports linking SERPINH1 to apoptosis and autophagy regulation [16,40]. Emerging evidence suggests that NETosis promotes tumorigenesis in hepatocellular carcinoma (HCC) by driving proliferation, metastasis, and EMT [41]. Conversely, alkaliptosis, a pH-dependent cell death mechanism [27,28], is emerging as a potential therapeutic target in cancer [42]. These observations position SERPINH1 as a multifunctional regulator of divergent cell death pathways. This paradigm-shifting discovery may inform the development of innovative therapeutic strategies targeting SERPINH1 or its downstream cell death effectors to suppress tumor growth and metastasis.

Finally, we identified PLOD1, ITGA5, and ESM1 as potential core collaborative genes with SERPINH1 that affect the prognosis of CESC patients. This molecular network not only plays a significant role in CESC but also shows similar prognostic predictive value in various tumor types, including KIRP, LGG, and MESO, suggesting the universality of its mechanism of action. Integrin-α5 (ITGA5), an N-glycoprotein, serves as a critical regulator of cellular adhesion capacity [43,44]. Procollagen-lysine,2-oxoglutarate 5-dioxygenase 1 (PLOD1) catalyzes the hydroxylation of lysine residues in procollagen

to facilitate collagen maturation and secretion [45], whereas SERPINH1 mediates proper folding of procollagen. These two molecules operate at distinct stages during collagen biosynthesis-to-maturation. Endothelial cell-specific molecule 1 (ESM1) plays pivotal roles in tumor angiogenesis [46]. These targets collectively form a functional cascade: cell adhesion, ECM remodeling, and angiogenesis. A deeper investigation into the interaction mechanisms between SERPINH1, PLOD1, ITGA5, and ESM1, and how they collectively influence tumor biological behavior, may reveal potential therapeutic targets for novel treatment strategies.

In summary, our study reveals that SERPINH1 promotes the malignant progression of CESC by enhancing the proliferation and invasion capabilities of CESC cells. Furthermore, GSVA analyses revealed associations between SERPINH1 expression and alterations in lipid metabolism, glycan metabolism pathways, and modulation of novel cell death modalities such as NETosis and Alkaliptosis. While these findings provide valuable insights into the multifaceted roles of SERPINH1, further experimental validation is necessary to confirm its regulatory effects on these pathways. Collectively, our study expands the understanding of SERPINH1 as a potential multi-pathway regulator in the malignant progression of CESC, highlighting its potential as a therapeutic target.

## Supporting information

**S1 Table. Comparative GSVA scores analysis of high- vs. low-SERPINH1 expression patient groups based on hallmark gene sets.**
(XLSX)

**S2 Table. Comparative GSVA scores analysis of high- vs. low-SERPINH1 expression patient groups based on Biocarta pathways.**
(XLSX)

**S3 Table. Comparative GSVA scores analysis of high- vs. low-SERPINH1 expression patient groups based on KEGG Legacy database.**
(XLSX)

**S4 Table. Comparative GSVA scores analysis of high- vs. low-SERPINH1 expression patient groups based on Fges.**
(XLSX)

**S5 Table. Comparative GSVA scores analysis of high- vs. low-SERPINH1 expression patient groups based on twelve cell death patterns.**
(XLSX)

**S1 File. Raw images.**
(PDF)

## Acknowledgments

We thank the Major Equipment Sharing Center of the Central South University for their technical support. We really appreciated Dr. Fa-Qing Tang for his critical editing of the article.

## Author contributions

**Conceptualization:** Qian Liu, Xiangjian Luo.

**Funding acquisition:** Xiangjian Luo.

**Supervision:** Wenbin Liu, Xiangjian Luo.

**Validation:** Yuanhao Peng.

**Visualization:** Qian Liu.

**Writing – original draft:** Qian Liu, Yuanhao Peng.

**Writing – review & editing:** Wenbin Liu, Xiangjian Luo.

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
