## [Decision Letter · Decision Letter 0]

PONE-D-25-09751SERPINH1 functions as a multifunctional regulator to promote the malignant progression of cervical cancerPLOS ONE

Dear Dr. Liu,

Thank you for submitting your manuscript to PLOS ONE. After careful consideration, we feel that it has merit but does not fully meet PLOS ONE’s publication criteria as it currently stands. Therefore, we invite you to submit a revised version of the manuscript that addresses the points raised during the review process. The authors must address the reviewers' comments, complete the introduction with appropriate references, justify the use of HeLa cells in their experiments, improve the quality of the figures, and enhance the discussion.

We look forward to receiving your revised manuscript.

Kind regards,

Victoria Pando-Robles, Ph.D.

Academic Editor

PLOS ONE

Journal Requirements:

2. Thank you for stating the following financial disclosure: [This work was supported by grants from the National Natural Science Foundation of China (82173144, 81874195).]. 

Reviewers' comments:

Reviewer's Responses to Questions

**Comments to the Author**

1. Is the manuscript technically sound, and do the data support the conclusions?

Reviewer #1: Yes

Reviewer #2: Partly

2. Has the statistical analysis been performed appropriately and rigorously? 

Reviewer #1: Yes

Reviewer #2: Yes

3. Have the authors made all data underlying the findings in their manuscript fully available?

Reviewer #1: Yes

Reviewer #2: Yes

4. Is the manuscript presented in an intelligible fashion and written in standard English?

Reviewer #1: Yes

Reviewer #2: Yes

5. Review Comments to the Author

Reviewer #1: The authors perform several bioinformatic analyses to show the importance of SERPINH1 in the promotion of cervical cancer malignancy. They also perform in vitro assays to support their findings. The manuscript is well written and has important findings. I consider it can be accepted after some minor comments are addressed.

Please specify how many times each experiment was repeated, either in the methods section or in the figure legends.

The subfigures of figure 4 are not correctly referred to in the text, please check.

The authors perform transfection experiments in SiHa and C33 cells, while they do shRNA experiments on HeLa cells. Why were these experiments performed on different cell ines? Please explain in the text.

The abstract states that This study systematically reveals the key role of SERPINH1 (Serpin Family H Member 1) as a hub regulator of malignant progression in cervical squamous cell carcinoma (CESC). However, HeLa cell line is an adenocarcinoma cell line and it is also used in this study. Please explain.

Section 6. Please define KIRP, LGG, MESO.

In the abstract, the authors mention the pathways uncovered by GSVA analysis (metabolism, netotic cell death and alkaliptosis). However, no experiments were perfomed to validate these findings. Although this is an important finding, this should not be mentioned in the abstract as one of the main findings in the study due to the lack of validation of the pathways.

As statistical analysis, the authors perform non-parametric analysis. Did the authors perform a normality test in their data in order to choose non-parametric tests? Please specify.

Reviewer #2: The manuscript presents a well-structured study with an appropriate methodological design. The topic is relevant, and the findings contribute to the understanding of SERPINH1 in cervical squamous cell carcinoma (CSCC). However, several issues should be addressed to strengthen the manuscript:

1. Lack of Supporting References in the Introduction:

The authors claim that the specific functions and regulatory mechanisms of SERPINH1 in cervical cancer remain poorly understood. This is an important point, but no references are provided to support this statement. Including relevant citations would substantiate the rationale for the study and contextualize its novelty.

2. Incomplete Literature Review in the Discussion:

While the discussion refers to some related studies, key prior work that could further support and enrich the interpretation of the findings is omitted. In particular, the study by Wang et al. (Front Genet., 2022; doi: 10.3389/fgene.2021.756094) offers valuable insights into the role of SERPINH1 across various cancers and should be included.

3. Low-Resolution Figures:

Figures 2, 3, and 6 are of suboptimal resolution, hindering proper evaluation of the results. The authors are encouraged to replace these with high-quality versions to ensure clarity and reproducibility.

4. Overstated Conclusions Regarding SERPINH1’s Role:

The manuscript proposes that SERPINH1 functions as a central regulator of malignant progression in CSCC via dual modulation of netotic cell death and alkaliptosis. However, the evidence presented does not conclusively support such a definitive role. The authors are advised to temper their conclusions and clearly distinguish between observed data and speculative interpretation.

Addressing these points will significantly enhance the clarity, scientific rigor, and impact of the manuscript.

6. PLOS authors have the option to publish the peer review history of their article (what does this mean? ). If published, this will include your full peer review and any attached files.

**Do you want your identity to be public for this peer review?** For information about this choice, including consent withdrawal, please see our Privacy Policy .

Reviewer #1: **Yes: ** Paola Maycotte

Reviewer #2: No

---

## [Author Response · Author response to Decision Letter 1]

20 May 2025

Point-by-Point Response

Reviewer #1: The authors perform several bioinformatic analyses to show the importance of SERPINH1 in the promotion of cervical cancer malignancy. They also perform in vitro assays to support their findings. The manuscript is well written and has important findings. I consider it can be accepted after some minor comments are addressed.

Response: Thank you for your careful review and positive comments! We are also grateful for your constructive suggestions, which we believe will significantly improve the clarity and impact of our manuscript.

Comment 1: Please specify how many times each experiment was repeated, either in the methods section or in the figure legends.

Response: Thanks for the reviewer pointing out this question. All experiments described in this study were independently repeated at least three times. To improve transparency and reproducibility, we have now explicitly stated the number of replicates in the revised ‘Methods’ section and have included this information in the corresponding figure legends.

Comment 2: The subfigures of figure 4 are not correctly referred to in the text, please check.

Response: Thank you for noticing this inconsistency. We have carefully revised the manuscript to ensure that the subfigures of Figure 4 are correctly referred to in the main text.

Comment 3: The authors perform transfection experiments in SiHa and C33 cells, while they do shRNA experiments on HeLa cells. Why were these experiments performed on different cell lines? Please explain in the text.

Response: We appreciate the reviewer's insightful question. We have added an explanation in the ‘Methods-Cell culture’ section of the manuscript. Briefly, the choice of different cell lines was based on their histological relevance and biological characteristics to best represent different aspects of cervical cancer. SiHa and C33A are representative of squamous cell carcinoma, while HeLa is an adenocarcinoma cell line, providing a broader understanding of SERPINH1’s effects. Additionally, our preliminary Western blot (WB) validation showed that SERPINH1 expression was relatively highest in HeLa cells, while it was lower in SiHa and C33A cells. This guided our decision to knock down SERPINH1 in HeLa cells and overexpress it in SiHa and C33 cells, thereby providing clearer insights into its role in tumor progression.

Comment 4: The abstract states that this study systematically reveals the key role of SERPINH1 as a hub regulator of malignant progression in cervical squamous cell carcinoma (CESC). However, HeLa cell line is an adenocarcinoma cell line and it is also used in this study. Please explain.

Response: We appreciate the reviewer for identifying this oversight. We apologize for the misunderstanding caused by the error. The full designation of CESC is Cervical Squamous Cell Carcinoma and Endocervical Adenocarcinoma, but it was incorrectly written as Cervical Squamous Cell Carcinoma (CESC) in the ‘Abstract’. This has been corrected in the revised manuscript. We sincerely thank you for catching this mistake. As previously mentioned, HeLa , SiHa, and C33A cell lines were selected to encompass the major histological subtypes of cervical cancer, thereby allowing a comprehensive assessment of SERPINH1's mechanistic role in cervical carcinogenesis.

Comment 5: Section 6. Please define KIRP, LGG, MESO.

Response: Thank you for catching this omission. The definitions for KIRP (kidney renal papillary cell carcinoma), LGG (brain lower grade glioma), and MESO (mesothelioma) have been added to Section 6 for clarity.

Comment 6: In the abstract, the authors mention the pathways uncovered by GSVA analysis (metabolism, netotic cell death, and alkaliptosis). However, no experiments were performed to validate these findings. Although this is an important finding, this should not be mentioned in the abstract as one of the main findings in the study due to the lack of validation of the pathways.

Response: We appreciate the reviewer’s insightful suggestion. We have revised the abstract to remove the reference to GSVA-derived pathways as main findings. Instead, these are now discussed as potential mechanisms warranting further experimental validation.

Comment 7: As statistical analysis, the authors perform non-parametric analysis. Did the authors perform a normality test in their data in order to choose non-parametric tests? Please specify.

Response: We thank the reviewer for this insightful comment. Our previous description of the statistical analysis methods was incomplete and has now been supplemented. Additionally, normality tests were indeed performed prior to the selection of non-parametric tests, and we have now explicitly mentioned this in the Statistical Analysis section of the manuscript: “Statistical analysis were analyzed using the R software (V.4.3.2, R Core Team, Foundation for Statistical Computing, Vienna, Austria) or GraphPad Prism 9 software (GraphPad Software Inc.). Student’s t-test was used for variables that met the requirements for normal distribution. The Wilcoxon signed-rank test was used for continuous variables that did not meet the requirement of normal distribution. Survival curves were estimated with the Kaplan-Meier method and subsequently compared using log-rank tests. P-value was set at p < 0.05 indicates significance. For all analysis, two-by-two pairs indicate statistically significant differences. *, **, *** and **** indicate, respectively <0.05, <0.01, <0.001, and <0.0001.”

Reviewer #2: The manuscript presents a well-structured study with an appropriate methodological design. The topic is relevant, and the findings contribute to the understanding of SERPINH1 in cervical squamous cell carcinoma (CSCC). However, several issues should be addressed to strengthen the manuscript.

Response: We appreciate your recognition of the manuscript's relevance and its potential contribution to the field of cervical cancer malignancy! We are also grateful for your constructive suggestions, which we believe will significantly enhance our manuscript.

Comment 1: Lack of Supporting References in the Introduction: The authors claim that the specific functions and regulatory mechanisms of SERPINH1 in cervical cancer remain poorly understood. This is an important point, but no references are provided to support this statement. Including relevant citations would substantiate the rationale for the study and contextualize its novelty.

Response: We appreciate the reviewer’s valuable suggestion. To strengthen the introduction, we have added relevant citations in paragraph 3 to support the claim that the specific functions and regulatory mechanisms of SERPINH1 in cervical cancer remain poorly understood. These references include recent studies that highlight gaps in current research and substantiate the novelty of our study.

Comment 2: Incomplete Literature Review in the Discussion:

Response: We appreciate the reviewer pointing out the need for a more comprehensive literature review. We have now incorporated the study by Wang et al. (Front Genet., 2022; doi: 10.3389/fgene.2021.756094), which provides significant insights into the role of SERPINH1 across various cancers: “Wang et al.'s pan-cancer analysis of SERPINH1 revealed its aberrant expression in 14 cancers, with high expression significantly reducing overall survival (OS), disease-specific survival, and progression-free interval in 11 cancers, highlighting its potential as a biomarker (10).” This addition helps contextualize our findings and supports the discussion on SERPINH1’s mechanisms in cervical cancer.

Comment 3: Low-Resolution Figures: Figures 2, 3, and 6 are of suboptimal resolution, hindering proper evaluation of the results. The authors are encouraged to replace these with high-quality versions to ensure clarity and reproducibility.

Response: Thanks for the reviewer pointing out this question. We have replaced Figures 2, 3, and 6 with high-resolution versions to improve clarity and facilitate proper evaluation.

Comment 4: Overstated Conclusions Regarding SERPINH1’s Role: The manuscript proposes that SERPINH1 functions as a central regulator of malignant progression in CSCC via dual modulation of netotic cell death and alkaliptosis. However, the evidence presented does not conclusively support such a definitive role. The authors are advised to temper their conclusions and clearly distinguish between observed data and speculative interpretation.

Response: We appreciate the reviewer’s valuable suggestion to refine our conclusions. In response, we have carefully revised the Abstract and Discussion sections to eliminate overstated claims and clearly distinguish between experimental observations and speculative interpretations. Conclusions regarding the regulatory role of SERPINH1 have been moderated to reflect the strength of the data presented. Specifically, the findings related to GSVA-derived pathways are now described as potential mechanisms that suggest biological relevance but warrant further experimental validation. This adjustment ensures that our interpretations remain aligned with the evidence and appropriately cautious: “In summary, our study reveals that SERPINH1 promotes the malignant progression of CESC by enhancing the proliferation and invasion capabilities of CESC cells. Furthermore, enrichment analyses suggest that SERPINH1 may influence tumor malignancy through the regulation of lipid metabolism reprogramming, activation of glycan metabolic pathways, and potential involvement in novel cell death modalities such as NETosis and Alkaliptosis. While these findings provide valuable insights into the multifaceted roles of SERPINH1, further experimental validation is necessary to confirm its regulatory effects on these pathways. Collectively, our study expands the understanding of SERPINH1 as a potential multi-pathway regulator in the malignant progression of CESC, highlighting its potential as a therapeutic target.”

---

## [Decision Letter · Decision Letter 1]

PONE-D-25-09751R1SERPINH1 functions as a multifunctional regulator to promote the malignant progression of cervical cancerPLOS ONE

Dear Dr. Liu,

Thank you for submitting your manuscript to PLOS ONE. After careful consideration, we feel that it has merit but does not fully meet PLOS ONE’s publication criteria as it currently stands. Therefore, we invite you to submit a revised version of the manuscript that addresses the points raised during the review process. The authors need to attended some reviewers comments before to accept the manuscript

We look forward to receiving your revised manuscript.

Kind regards,

Victoria Pando-Robles, Ph.D.

Academic Editor

PLOS ONE

Journal Requirements:

Reviewers' comments:

Reviewer's Responses to Questions

**Comments to the Author**

1. If the authors have adequately addressed your comments raised in a previous round of review and you feel that this manuscript is now acceptable for publication, you may indicate that here to bypass the “Comments to the Author” section, enter your conflict of interest statement in the “Confidential to Editor” section, and submit your "Accept" recommendation.

Reviewer #1: All comments have been addressed

Reviewer #3: (No Response)

2. Is the manuscript technically sound, and do the data support the conclusions?

Reviewer #1: Yes

Reviewer #3: Partly

3. Has the statistical analysis been performed appropriately and rigorously? 

Reviewer #1: Yes

Reviewer #3: No

4. Have the authors made all data underlying the findings in their manuscript fully available?

Reviewer #1: Yes

Reviewer #3: Yes

5. Is the manuscript presented in an intelligible fashion and written in standard English?

Reviewer #1: Yes

Reviewer #3: Yes

6. Review Comments to the Author

Reviewer #1: The authors have answered my previous comments and I have no further comments. I recommend the manuscript for publication.

Reviewer #3: In this research article, the authors aim to characterize the role of SERPINH1 in the progression of cervical squamous cell carcinoma (CESC), using several bioinformatics analyses and complemented by in vitro experiments. Through those analyses, it was identified that CC patients with high expression of SERPINH1 showed stromal remodeling, enhanced angiogenesis, and lipid metabolic alterations. However, the main results observed were that SERPINH1 is involved in proliferation, migration, and invasion pathways, which were identified in silico and corroborated with in vitro experiments.

This is an interesting and well-written research that is looking for the SERPINH1 role in CESC, but there are several points that need to be addressed.

General comment

It is not clear why the authors concluded that SERPINH1 is involved in the regulation of lipid metabolism. The results presented here showed that SERPINH1 is overexpressed, and several alterations could be related to this protein, but not necessarily in a direct effect. With the results obtained in silico, the authors cannot explain the mechanism through which SERPINH1 modulates this pathway. The authors are overestimating the results, and because of that, the discussion section needs to be modified.

In the in vitro assays, the authors need to better characterize the cell lines for SERPINH1 expression after being transduced or transfected to modulate SERPINH1 expression.

The cell lines used in the experiments were SiHa, which is a cervical cancer cell line that is transformed with HPV16; C33 is a cervical cancer cell line that does not contain HPV; and HeLa, which is an adenocarcinoma cell line transformed with HPV18. It will be important that the authors justify the use of different cell lines in the experiments and how the different genetic and viral backgrounds of the cell lines could affect the results.

An interesting result is presented in Fig. 6I, where the authors showed the regulation of the RNA expression levels of ITGA5, PLOD1, and ESM1 by SERPINH1. These results should be better discussed, as this is an event where SERPINH1 expression levels were directly involved and influenced the expression of these genes.

Specific points

Lines 141-145. Determine statistical differences between groups. Here, it is not clear which test was used, as for parametric data the t-test is used, but for non-parametric data the Mann-Whitney U is used. Please clarify and add this information to this section.

Line 224. Here it should be Fig. 4B and C

Line 226. The figure reference is wrong; this should be 4D and E

Line 228. Here again, the figure reference is wrong as it should be 4F-I

Lines 229-238. The authors suggested that SERPINH1 is overexpressed in CC and that the overexpression of this protein is related to the promotion of invasion and metastasis. To corroborate these findings that were identified from different databases, the researchers used cell lines that came from different cervical cancers. The researchers transduced SiHa and C33 to overexpress SERPINH1, but the basal protein levels seemed to be high. When the cell lines were transduced, no differences in SERPINH1 levels were observed (Fig. 4A). It would be desirable to carry out a densitometric analysis of this data and to have a quantitative analysis to identify the differences between the vector and the transduced SERPINH1.

On the other hand, the researchers used the HeLa cell line (not transduced), and it is clear from the Western blot that the basal levels of SERPINH1 were high in this cell line (Fig.5A). Why did the authors not use this cell line in all the experiments? How do the basal levels of SERPHIN1 affect or not the results observed?

Also, it would be ideal to determine the percentage of inhibition of SERPINH1 expression after siRNA transfection (Fig. 5A). This is important because siSERPINH1#1 seems to express lower levels of this protein than siSERPINH1#2, and this has different effects on the colony formation, migration, and invasion assays. Revise this and add this information to the text and in the corresponding figure to strengthen the analysis of the results.

Lines 231-238. In this part of the experiments to determine the role of SERPINH1 in proliferation, migration, and invasion by knocking down the gene, the researchers used the Hela cells. However, for the previous part of the experiments to show that overexpression stimulates these pathways, the cell lines used were SiHa and C33. There is no explanation for this change. The authors need to address the change of cell lines in the experiments and justify the use of the different cell lines.

Lines 302-306. The authors concluded that “SERPINH1 promotes the malignant progression of CESC by enhancing the proliferation and invasion capabilities of these cells”. These are the facts that were tested under in vitro conditions and that could be accepted to be associated with SERPINH1.

However, in this other part, “SERPINH1 in tumor malignancy through the regulation of lipid metabolism reprogramming, activation of glycan metabolic pathways, and modulation of novel cell death modalities such as NETosis and Alkaliptosis”, the authors are speculating too much about the role of SERPINH1 in regulating these pathways, as this analysis was made through analysis of patients bank genes. To confirm the role of SERPINH1 in these pathways, in vitro experiments are needed. The authors may construct an interaction diagram, which can better explain the interactions between these pathways and SERPINH1, and identify what needs to be done next.

7. PLOS authors have the option to publish the peer review history of their article (what does this mean? ). If published, this will include your full peer review and any attached files.

**Do you want your identity to be public for this peer review?** For information about this choice, including consent withdrawal, please see our Privacy Policy .

Reviewer #1: **Yes: ** Paola Maycotte

Reviewer #3: No

---

## [Author Response · Author response to Decision Letter 2]

6 Jul 2025

Point-by-Point Response

Reviewer #1: The authors have answered my previous comments and I have no further comments. I recommend the manuscript for publication.

Response: We are deeply grateful for your time and expertise in reviewing our manuscript! Your constructive feedback during the initial review significantly strengthened our study. Thank you for your final endorsement—we truly appreciate your support in advancing this work.

Reviewer #3: This is an interesting and well-written research that is looking for the SERPINH1 role in CESC.

Response: Thank you for your careful review and positive comments! We are also grateful for your constructive suggestions, which we believe will significantly improve the clarity and impact of our manuscript.

Comment 1:It is not clear why the authors concluded that SERPINH1 is involved in the regulation of lipid metabolism. The authors are overestimating the results, and because of that, the discussion section needs to be modified.

Response: We appreciate the reviewer's insightful question. Our claim was based on the following bioinformatic evidence (now explicitly framed as preliminary association in the revised text): In TCGA-CESC cohort analysis (n=304), low SERPINH1 expression tumors showed significant enrichment in lipid metabolism pathways (e.g., LINOLEIC_ACID_METABOLISM: LogFC = -0.62, p<0.000) compared to high SERPINH1 group.

We have revised the ‘Discussion’: These findings revealed an inverse correlation between SERPINH1 expression and activity scores of lipid metabolic pathways. While this suggests a potential link, direct mechanistic evidence for SERPINH1-driven lipid reprogramming requires experimental validation. The clarification of this mechanism could offer novel molecular targets for the development of specific therapeutic strategies aimed at combating metastatic CESC.

Comment 2:An interesting result is presented in Fig. 6I, where the authors showed the regulation of the RNA expression levels of ITGA5, PLOD1, and ESM1 by SERPINH1. These results should be better discussed, as this is an event where SERPINH1 expression levels were directly involved and influenced the expression of these genes.

Response: We thank the reviewer for highlighting the significance of these findings. As suggested, we have now expanded the discussion of ITGA5, PLOD1, and ESM1 in the revised manuscript (The fourth paragraph of Discussion).

Comment 3: Lines 141-145. Determine statistical differences between groups. Here, it is not clear which test was used, as for parametric data the t-test is used, but for non-parametric data the Mann-Whitney U is used. Please clarify and add this information to this section.

Response: We thank the reviewer for emphasizing the need for methodological transparency. The statistical analyses have now been explicitly specified in the revised ‘Methods’ section.

Comment 4: Line 224. Here it should be Fig. 4B and C

Line 226. The figure reference is wrong; this should be 4D and E

Line 228. Here again, the figure reference is wrong as it should be 4F-I

Response: Thank you for carefully checking the figure citations. We have re-examined the original manuscript and confirm that the figure references at these locations were correctly denoted as follows in our submitted version:

Line 224: (Fig 4B-C) revised to (Fig. 4B and 4C)

Line 226: (Fig 4D-E) revised to (Fig. 4D and 4E)

Line 228: (Fig 4F-I) revised to (Fig. 4F-I)

For other figure references in this manuscript, we have also made corresponding modifications to ensure the uniform format.

Comment 5: The researchers transduced SiHa and C33 to overexpress SERPINH1, but the basal protein levels seemed to be high. When the cell lines were transduced, no differences in SERPINH1 levels were observed (Fig. 4A). It would be desirable to carry out a densitometric analysis of this data and to have a quantitative analysis to identify the differences between the vector and the transduced SERPINH1. Also, it would be ideal to determine the percentage of inhibition of SERPINH1 expression after siRNA transfection (Fig. 5A). This is important because siSERPINH1#1 seems to express lower levels of this protein than siSERPINH1#2, and this has different effects on the colony formation, migration, and invasion assays. Revise this and add this information to the text and in the corresponding figure to strengthen the analysis of the results.

Response: Thank you for your valuable feedback, which is highly significant for improving our manuscript. To address your concerns, we repeatedly overexpressed SERPINH1 in both Siha and C33A cell lines. The expression levels were detected by Western blotting (WB), and in accordance with your suggestions, we performed quantitative analysis of the WB bands. The results demonstrated that SERPINH1 was upregulated approximately 6-fold in the Siha cell line and approximately 4-fold in the C33A cell line. These data clearly demonstrate a significant difference between the vector control group and the SERPINH1-transfected group (Figure 4A).

In addition, we have quantified the WB bands in Figure 5A and included the percentage inhibition of SERPINH1 expression levels. As shown in the figure, the protein expression level achieved with siSERPINH1#1 was slightly lower than that with siSERPINH1#2, although no statistically significant difference was observed between the two. Furthermore, in subsequent biological functional assays, slight differences between siSERPINH1#1 and siSERPINH1#2 may have arisen due to batch-to-batch variations in transfection efficiency. However, these observed differences fell within statistically acceptable error margins and, crucially, do not affect our conclusion that SERPINH1 promotes the proliferation, migration, and invasion of cervical cancer cells.

Revised Fig. 4A

Revised Fig. 5A

The new Fig. 4A and Fig. 5A have been added to the revised manuscript. We apologize for omitting this key quantification in the original submission.

Comment 6: In this part of the experiments to determine the role of SERPINH1 in proliferation, migration, and invasion by knocking down the gene, the researchers used the Hela cells. However, for the previous part of the experiments to show that overexpression stimulates these pathways, the cell lines used were SiHa and C33. There is no explanation for this change. The authors need to address the change of cell lines in the experiments and justify the use of the different cell lines.

On the other hand, the researchers used the HeLa cell line (not transduced), and it is clear from the Western blot that the basal levels of SERPINH1 were high in this cell line (Fig.5A). Why did the authors not use this cell line in all the experiments? How do the basal levels of SERPHIN1 affect or not the results observed?

Response: We appreciate the reviewer's question. We have an explanation in the ‘Methods-Cell culture’ section of the manuscript. Briefly, the choice of different cell lines was based on their histological relevance and biological characteristics to best represent different aspects of cervical cancer. SiHa and C33A are representative of squamous cell carcinoma, while HeLa is an adenocarcinoma cell line, providing a broader understanding of SERPINH1’s effects. Additionally, our preliminary Western blot (WB) validation showed that SERPINH1 expression was relatively highest in HeLa cells, while it was lower in SiHa and C33A cells. This guided our decision to knock down SERPINH1 in HeLa cells and overexpress it in SiHa and C33 cells, aiming to induce greater variations in SERPINH1 expression levels within these cell lines, thereby highlighting the impact of this molecule on cancer cells.

Comment 7:“SERPINH1 in tumor malignancy through the regulation of lipid metabolism reprogramming, activation of glycan metabolic pathways, and modulation of novel cell death modalities such as NETosis and Alkaliptosis”, the authors are speculating too much about the role of SERPINH1 in regulating these pathways, as this analysis was made through analysis of patients bank genes.

Response: We appreciate the reviewer’s insightful suggestion. We have revised the discussion to remove the reference to GSVA-derived pathways as main findings. Instead, these are now discussed as potential mechanisms warranting further experimental validation: Furthermore, GSVA analyses revealed associations between SERPINH1 expression and alterations in lipid metabolism, glycan metabolism pathways, and modulation of novel cell death modalities such as NETosis and Alkaliptosis. While these findings provide valuable insights into the multifaceted roles of SERPINH1, further experimental validation is necessary to confirm its regulatory effects on these pathways.

---

## [Editor Report · Decision Letter 2]

SERPINH1 functions as a multifunctional regulator to promote the malignant progression of cervical cancer

PONE-D-25-09751R2

Dear Dr. Liu,

We’re pleased to inform you that your manuscript has been judged scientifically suitable for publication and will be formally accepted for publication once it meets all outstanding technical requirements.

Kind regards,

Victoria Pando-Robles, Ph.D.

Academic Editor

PLOS ONE
---

## [Editor Report · Acceptance letter]

PONE-D-25-09751R2

PLOS ONE

Dear Dr. Liu,

I'm pleased to inform you that your manuscript has been deemed suitable for publication in PLOS ONE. Congratulations! Your manuscript is now being handed over to our production team.

Kind regards,

on behalf of

Dr. Victoria Pando-Robles

Academic Editor

PLOS ONE